# Oviposition Preference and Developmental Performance of *Drosophila suzukii* on Different Cherry Cultivars

**DOI:** 10.3390/insects15120984

**Published:** 2024-12-11

**Authors:** Fan Yang, Haikuan Sun, Zehua Wang, Jingxia Xie, Jingyan He, Guanghang Qiao, Jing Wang, Yuyu Wang, Shanning Wang

**Affiliations:** 1Key Laboratory of Environment Friendly Management on Fruit and Vegetable Pests in North China (Co-Construction by Ministry and Province), Ministry of Agriculture and Rural Affairs, Beijing 100093, China; evelynyangfan@163.com (F.Y.); 13114869707@163.com (H.S.); wangzehua200707@163.com (Z.W.); 18289840755@139.com (J.X.); h2768354134@163.com (J.H.); qghang98@126.com (G.Q.); sduwj@126.com (J.W.); 2Institute of Plant Protection, Beijing Academy of Agriculture and Forestry Sciences, Beijing 100093, China; 3College of Plant Protection, Hebei Agricultural University, Baoding 071000, China; wangyy_amy@126.com; 4Institute of Forestry and Pomology, Beijing Academy of Agriculture and Forestry Sciences, Beijing 100093, China

**Keywords:** spotted-wing drosophila, sweet cherry, oviposition preference, growth and development parameters, fruit physiological parameters, pest control

## Abstract

*Drosophila suzukii*, a major pest in cherry orchards, shows distinct oviposition preferences influenced by cherry cultivar traits such as the color, firmness, and sugar content. In this study, *D. suzukii* laid the highest number of eggs on fruits of the “Hongdeng” and “Burlat” cherry cultivars, and the number of eggs laid on fruits of other cultivars significantly increased with the fruit maturity. These preferences correlated with the fruit characteristics, as darker-colored cherries consistently received more eggs. The developmental parameters of *D. suzukii* varied among the cultivars, with “Hongdeng” cherries supporting the highest pupation rate and survival rate. The analysis further showed that darker color parameters (lightness (*L**) and chromaticity (*b**)) were negatively correlated with the oviposition, while the *CIRG* (color index for red grapes) values were positively associated with increased egg-laying and pupation rates. The findings confirmed the common notion that “rotting fruit attracts flies”, and *D. suzukii* is no exception. The results also demonstrate that the fruit color plays a role in the host selection process of *D. suzukii*, providing valuable insights for developing targeted pest management strategies in cherry production.

## 1. Introduction

*Drosophila suzukii* (Mastumura), commonly known as the spotted-wing drosophila, is a major fruit pest worldwide. It possesses a well-developed, serrated ovipositor, which allows it to lay eggs in unripe fruit, causing severe damage to soft-skinned stone fruits such as cherries [1,2] and leading to significant economic losses [3,4,5]. Sweet cherry (*Prunus avium* L.), belonging to the Rosaceae family, is one of the most popular temperate fruits and has great economic importance [6,7,8]. In recent years, China has emerged as the largest consumer of sweet cherries globally, as highlighted by the USDA’s annual report. By 2024/2025, the Chinese cherry production reached 850,000 tons, with a planting area of approximately 199,000 hectares, ranking first in the world [9]. Sweet cherries are among the key economic fruit crops in China, playing a vital role in enhancing agricultural efficiency, increasing farmers’ income, and promoting local economic development [10]. The earliest report of *D. suzukii* infestation in sweet cherries was in California, USA, in 2009 [11], while in China, the first occurrence was reported in 2013 in late-maturing sweet cherries in Shandong Province [12]. Currently, the pest is widespread across the major cherry-producing regions in China, severely affecting the industry’s development [13,14].

The oviposition preference of *D. suzukii* is influenced by various fruit characteristics, including the color [15,16,17], firmness [18,19], nutrients [19], and ripeness [20]. Studies have shown that *D. suzukii* can assess the suitability of different fruits as hosts based on the color reflection of the fruit [16]. Field experiments have revealed that *D. suzukii* tends to lay eggs on grapes with softer skins, and indoor tests on six types of berries from vineyards also confirmed this preference [18]. Additionally, although fruit nutrients, such as acids, can affect the development of the larvae, they do not have a significant impact on the hatching of adults [19]. *Drosophila suzukii* is particularly attracted to overripe strawberries, and its attraction increases as the fruit ripens [20]. Further research has shown that *D. suzukii* exhibits significant differences in its oviposition preferences among different cherry cultivars, favoring those with higher glycogen contents, lower pectin contents, and softer flesh [21].

China’s sweet cherry industry boasts remarkable genetic diversity, with over 90 cultivars currently grown, including both internationally renowned and domestically bred varieties. Cultivars such as “Tieton” (USA), “Burlat” (France), and “Rainier” (USA) are widely cultivated and highly regarded for their quality. Domestically bred cultivars have also gained prominence in recent decades. For example, “Hongdeng”, developed by the Institute of Pomology at the Dalian Academy of Agricultural Sciences (DAAS), was the first-generation main cultivar in China and played a pivotal role in the early development of the country’s cherry industry. Additionally, “Caihong”, bred by the Institute of Pomology and Forestry at the Beijing Academy of Agriculture and Forestry Sciences (BAAFS), is another widely cultivated variety. More recently, “Xiangquan No. 1”, also developed by the BAAFS, became the first self-fertile sweet cherry cultivar released in China, marking a significant breakthrough in breeding for self-fertility [22].

In practical production, cherry orchards often grow multiple cultivars, which show differences in their physiological and phenological characteristics [23]. The relationship between these differences among cultivars and the oviposition preference, growth, and development of *D. suzukii* require further investigation. Therefore, we hypothesize that *D. suzukii* exhibits differential oviposition preferences among different cherry cultivars, favoring more mature and higher-quality fruits, which may provide better conditions for the growth and development of its offspring. To test this hypothesis, we selected six major cherry cultivars with a gradient of physiological characteristics. In this study, we aimed to (1) measure the physiological traits of different cherry cultivars, including the fruit color, firmness, and sugar content; (2) evaluate the oviposition preferences of *D. suzukii* across these cherry cultivars; and (3) investigate the effects of different cherry cultivars on the growth, development, and emergence of *D. suzukii* and analyze the correlations among the oviposition preference, developmental parameters, and physiological characteristics of cherry fruits. We further aimed to elucidate the mechanisms underlying the *D. suzukii* oviposition behavior, providing a theoretical foundation for breeding cherry cultivars with pest resistance, and offering scientific guidance for integrated pest management in cherry production to effectively address the threat posed by *D. suzukii*.

## 2. Materials and Methods

### 2.1. Insect Colony

The experimental insect population was a laboratory strain of *D. suzukii*, reared since 2022, whose diet formulation was based on the recipe described by Liu et al. [24], with a minor modification: the replacement of 2 mL of the vitamin mixture with 2 g of insect vitamin, specifically the Vanderzant vitamin mixture for insects (catalog No. V1007-100G; Sigma-Aldrich, St. Louis, MO, USA). The diet mainly consisted of wheat bran, sucrose, brewer’s yeast powder, preservatives, and vitamins. Rearing and experimental conditions included maintaining at 24 ± 1 °C, a photoperiod of L:D = 16 h:8 h, and 70 ± 5% relative humidity. Adults aged 1–5 days post-eclosion from the same cohort were selected for the subsequent oviposition preference trials.

### 2.2. Test Fruits

The test cherry fruits were collected from a cherry orchard in Tongzhou District, Beijing (116°42′ E, 39°41′ N). Based on differences in the fruit color and ripeness time, six cherry cultivars were selected: “Hongdeng” (HD), “Tieton” (TT), “Burlat” (BLT), “Xiangquan No.1” (XQ1), “Rainier” (RN), and “Caihong” (CH). Among them, HD, TT, and BLT are early-ripening cultivars, while XQ1, RN, and CH ripen later. From 16 May to 12 June 2024, fruits were collected weekly for a total of five harvests. Due to the differences in the ripeness periods, the number of collection operations varied among the cultivars, and the collection times were categorized into the ripe and overripe stages (see Appendix A). For each cultivar, fruits were randomly harvested from two rows, with approximately 15 trees per row, ensuring an adequate number of fruits for the experiments. After collection, the fruits were brought to the laboratory, where those with uniform size, clean surfaces, and no signs of pest damage or mechanical injury were selected for the physiological measurements and subsequent experiments. All cherry cultivars were cultivated in the same orchard, with trees approximately 10 years old. Standard agricultural practices were followed, and no insecticides were used before or during fruit maturation. Lime sulfur was applied before bud break in early spring, and mancozeb was sprayed during full bloom.

### 2.3. Measurement of Cherry Fruit Color, Firmness, and Sugar Content

#### 2.3.1. Fruit Color

The color of the cherry fruits was measured by using an LS173 colorimeter (Linshang Technology Co., Ltd., Shenzhen, China). To ensure consistency and minimize measurement error, all measurements were conducted by the same operator. For each cherry cultivar, 30 fruits with uniform ripeness were selected, and color measurements were taken at three points on the fruit: the equator, the top, and the bottom. At each ach position, measurements were taken three times, and the average value was used as the color parameter for the fruit (lightness (*L**) and chromaticity (*a** and *b**) values). *L** represents lightness, ranging from 0 (black) to 100 (white). *a** indicates the red–green axis, with positive values toward red and negative values toward green. *b** denotes the yellow–blue axis, with positive values toward yellow and negative values toward blue [25,26]. From these, the color saturation [*C** = (*a*^2^ + *b*^2^)^1/2^], hue angle [h° = arctan(*b**/*a**)] [27], and color index of red grapes (*CIRG*) [*CIRG* = (180 h°)/(*L** + *C**)] [28,29] were calculated. The fruit color was classified based on the *CIRG* value as follows: *CIRG* < 2, yellow-green; 2 < *CIRG* < 4, pink; 4 < *CIRG* < 5, red; 5 < *CIRG* < 6, dark red; *CIRG* > 6, blue-black [30].

#### 2.3.2. Fruit Firmness

The fruit firmness was measured by using a GY-4 analog fruit firmness tester (Aidebao Instrument Co., Ltd., Leqing, China). To ensure data accuracy, all measurements were conducted by the same operator. An 8 mm diameter probe was used with a penetration depth of 5 mm, and the maximum value recorded was taken as the firmness of the fruit. For each cultivar, 30 fruits of uniform size were selected, and the firmness was measured at the top, middle, and bottom of the fruit. The hardest area among the three was chosen as the firmness value for that fruit.

#### 2.3.3. Fruit Sugar Content

The sugar content of the cherry fruits was measured by using a PAL-1 refractometer (ATAGO Scientific Instruments Co., Ltd., Guangzhou, China). To minimize human error, all measurements were conducted by the same operator. For each cultivar, 30 fruits of uniform size were selected, and the flesh from the equatorial region of each fruit was blended into a homogenate. The juice was then filtered through gauze, and a thin layer of the juice was evenly applied to the PAL-1 refractometer’s prism for measurement. Each fruit’s sugar content was measured three times, and the average value was recorded as the sugar content of the fruit. The unit for sugar content data is °Brix. °Brix, or °Bx, is a unit used to measure the soluble solid content in a solution, primarily for assessing the sugar contents in fruits, juices, and products such as wine. One degree Brix is 1 g of sucrose in 100 g of solution (1°Brix = 1% sugar) [31].

### 2.4. Oviposition Preference Test of Drosophila suzukii on Fruits of Different Cherry Cultivars

#### 2.4.1. Non-Choice Oviposition

Six whole cherry fruits of different cultivars were individually placed into custom-made insect rearing boxes (volume of 450 mL; top diameter of 11 cm; bottom diameter of 8.5 cm; height of 7 cm; catalog No. 12189879065; Labahua, Xiamen, China). A hole was made in the center of each box lid and covered with 120-mesh gauze to ensure ventilation. Ten adult *D. suzukii* flies, aged 1 to 5 days (sex ratio of 1:1), were introduced into each box for mating and oviposition. The boxes were kept in an artificial climate chamber (model RXZ-328A; Ningbo Jiangnan Instrument Factory, Ningbo, China) under controlled conditions: a temperature of 24 ± 1 °C, a photoperiod of L:D = 16 h:8 h, and a relative humidity of 70 ± 5%. After 24 h, the adult flies were removed, and each fruit was examined with a stereomicroscope (SZ51; Olympus Corporation, Tokyo, Japan) to assess the oviposition. The number of eggs laid on each fruit was recorded. Each cherry cultivar had eight replicates.

#### 2.4.2. Choice Oviposition

The choice oviposition test was conducted only during the overripe stage, following a similar procedure to that of the non-choice oviposition test. Cherry fruits of different cultivars were placed in custom-made insect rearing boxes with their lids kept open. The number of cultivars used depended on the fruit collection availability on the day (see Appendix A). The rearing boxes were then placed inside the insect rearing cages (dimensions: 20 cm × 20 cm × 20 cm; AIPU INSTRUMENT; Hangzhou, China), where 40–60 adult *D. suzukii* flies (sex ratio of 1:1) were introduced. The cages were kept in an artificial climate chamber under the same conditions as those of the non-choice oviposition test. After 24 h, the adult flies were removed, and the number of eggs on each fruit was counted under a stereomicroscope. Each cultivar had eight replicates, and the fruits from different cultivars were randomly positioned within the cage.

### 2.5. Measurement of Growth and Development Parameters of Drosophila suzukii on Different Cherry Cultivars

#### 2.5.1. Development Periods

In the overripe stage of the cherries, fully mature and pest-free fruits from different cultivars were selected for the experiment. The cherry fruits were placed in a *D. suzukii* adult rearing cage (dimensions: 35 cm × 35 cm × 35 cm; AIPU INSTRUMENT; Hangzhou, China), and after 24 h, the fruits were removed to ensure that the adult flies had sufficient time to lay eggs. The hatching of larvae was observed and recorded daily, and newly hatched larvae were individually transferred into a 24-well cell culture plate (REF 3524, Corining Incorporated, Kennebunk, ME, USA), where they were fed with the flesh of the same cherry cultivar. The egg-to-larva period (Egg–Larva Period) and the pupal developmental period (Pupal Period) were recorded. Four replicate groups were set up for each cherry cultivar, with 24 larvae in each group. The experiment was conducted in an artificial climate chamber under the same environmental conditions as previously described.

#### 2.5.2. Other Growth and Development Parameters

The six cherry cultivars were placed in separate *D. suzukii* rearing cages (dimensions: 35 cm × 35 cm × 35 cm; AIPU INSTRUMENT; Hangzhou, China) for 24 h to allow for oviposition. Afterward, the fruits were removed, and the number of eggs laid on the fruits of each cultivar was counted to ensure that each had approximately 150 eggs. Any excess eggs were carefully removed by using sterilized insect pins. Four replicate experiments were conducted for each cultivar. The cherry fruits (approximately four fruits per cultivar) were then individually placed in insect rearing boxes and kept in an artificial climate chamber under the same conditions as previously described. The numbers of larvae hatched, pupae formed, and emerged adults and the sex ratio of the adults were recorded daily. Based on these observations, the following parameters were calculated: hatching rate = number of larvae hatched/total number of eggs; pupation rate = number of pupae/total number of larvae; eclosion rate = number of emerged adults/total number of pupae; survival rate = number of emerged adults/total number of eggs; female ratio = number of female adults/total number of adults.

### 2.6. Data Analysis

All data analyses were performed with R 4.4.0 software. First, a generalized linear model (GLM) was used to analyze the effects of the cultivar, ripeness stage, and their interaction on the physiological parameters of the cherries, with *t*-tests conducted to compare the ripe and overripe stages for the same parameter. The same approach was applied to the non-choice oviposition data, while the choice oviposition data, which did not meet the assumption of the homogeneity of variance, were analyzed by using non-parametric tests. For the developmental parameters of *D. suzukii*, the normality and homogeneity of variance were first tested; if the assumptions were met, then a one-way ANOVA followed by Tukey’s HSD post hoc tests were used; otherwise, Kruskal–Wallis tests and Dunn’s test were applied. Pearson correlation coefficients were calculated by using the cor() function to assess the linear relationships between the variables, and a correlation matrix was generated. Bar plots and linear regression plots were created by using the ggplot2 package.

## 3. Results

### 3.1. Physiological Parameters of Different Cherry Cultivars

The physiological parameters of the cherry fruits, including the color (*L**, *a**, and *b**), firmness, and sweetness, showed significant differences between the cultivars and across the different ripeness stages (ripe and overripe stages) (*p* < 0.001; see Appendix A). The color parameters (*L**, *a**, and *b**), which were used to calculate the color index for red grapes (*CIRG*), generally had higher average values in the ripe stage than in the overripe stage, with the exception of *a** in XQ1 and CH and *b** in RN (see Appendix A). As the fruit matured, the cherry fruit’s color (*CIRG*) deepened, and the firmness decreased. Among the cultivars, BLT consistently had the highest *CIRG* value and the lowest firmness. In terms of the sugar content, with the exception of XQ1 having a higher value in the ripe stage, the other cultivars all showed higher sugar contents in the overripe stage. Notably, HD and BLT exhibited the highest sugar contents throughout the entire period (see Appendix A and Figure 1).

### 3.2. Oviposition of Drosophila suzukii on Different Cherry Cultivars

#### 3.2.1. Non-Choice Oviposition

In the non-choice oviposition experiment, *D. suzukii* exhibited significant differences in its oviposition on fruits of different cherry cultivars (*F*_5, 162_ = 39.45, *p* < 0.001) and in different ripeness stages (F_1, 161_ = 32.29, *p* < 0.001). Specifically, there was no significant difference in the oviposition on the HD and BLT cultivar fruits between the ripe and overripe stages, whereas for the other cultivars, the oviposition was significantly higher during the overripe stage than during the ripe stage (see Appendix A, Figure 2). In the ripe stage, the highest average oviposition was observed on the HD cherries, with an average of 32.0 ± 4.8 eggs per cherry, followed by BLT, TT, XQ1, RN, and CH, with CH having the lowest average oviposition at 4.1 ± 0.6 eggs per cherry. In the overripe stage, the HD cherries again had the highest average oviposition at 31.9 ± 2.0 eggs per cherry, followed by BLT, XQ1, TT, RN, and CH, where CH had the lowest average oviposition at 12.1 ± 1.3 eggs per cherry (see Appendix A, Figure 2). The data are presented as average values ± standard errors (SDs). Throughout the entire period, the oviposition was consistently the highest on the HD and BLT cultivar fruits, while it was the lowest on the RN and CH cultivar fruits (see Figure 2).

#### 3.2.2. Choice Oviposition

The choice oviposition of *D. suzukii* on fruits of different cherry cultivars during the overripe stage was compared, and the results show significant differences among the cultivars (χ^2^ = 29.10, df = 5, *p* < 0.001). The average number of eggs laid, from the highest to the lowest, was as follows: HD (32.4 ± 3.2 eggs), BLT (27.6 ± 1.9 eggs), XQ1 (22.44 ± 3.212 eggs), TT (22.4 ± 1.6 eggs), CH (15.3 ± 2.2 eggs), and RN (14.5 ± 2.3 eggs). The average oviposition on the HD and BLT fruits was significantly higher than that on the CH and RN fruits (see Figure 3).

### 3.3. Growth and Development Parameters of Drosophila suzukii on Fruits of Different Cherry Cultivars

For assessing the developmental time of *D. suzukii*, the Egg–Larva Period and Pupal Period were recorded. There were significant differences in the Egg–Larva Period among the different cherry cultivars (χ^2^ = 41.07, df = 5, *p* < 0.001), with the average development duration ranging from 4.82 to 4.98 d. The order from the longest to the shortest was HD > TT > XQ1 > BLT > CH > RN. The average Pupal Period ranged from 4.70 to 4.88 d, with RN having the longest duration and XQ1 the shortest. Additionally, the hatching, pupation, eclosion, and survival rates were calculated. Both the pupation rate (F_5, 42_ = 2.743, *p* = 0.031) and survival rate (F_5, 42_ = 2.658, *p* = 0.036) showed significant differences among the cultivars. The ranking of the pupation rate from the highest to the lowest was HD > BLT > TT > XQ1 > RN > CH, while for the survival rate, the order was HD > BLT > XQ1 > TT > CH > RN. The average hatching rates and eclosion rates ranged from 0.83 to 0.89 and from 0.74 to 0.81, respectively (see Table 1). The average female ratio ranged from 0.58 to 0.67, with HD and BLT showing lower values, both below 0.60 (see Table 1).

### 3.4. Correlation Analysis of Choice Oviposition, Developmental Parameters, and Fruit Physiological Parameters

The choice oviposition showed a significant negative correlation with the color parameters *L** (r = −0.85, *p* = 0.030) and *b** (r = −0.89, *p* = 0.019), and a marginally significant positive correlation with the *CIRG* (r = 0.80, *p* = 0.055) (see Table 2, Figure 4). The choice oviposition was negatively correlated with the fruit firmness and positively correlated with the sugar content; however, these correlations were not significant (see Table 2).

The Egg–Larva Period also exhibited a significant negative correlation with the *L** (r = −0.88, *p* = 0.020), and the hatching rate was also significantly negatively correlated with the *L** (r = −0.82, *p* = 0.047). Additionally, the pupation rate was significantly negatively correlated with the b (r = −0.82, *p* = 0.044) and positively correlated with the *CIRG* (r = 0.84, *p* = 0.035) (see Table 2, Figure 5). The Egg–Larva Period was negatively correlated with the firmness and positively correlated with the sugar content; the Pupal Period showed positive correlations with both the firmness and sugar content. The female ratio was positively correlated with the firmness and negatively correlated with the sugar content. Additionally, the hatching, pupation, and eclosion rates were all negatively correlated with the firmness and positively correlated with the sugar content. However, none of these correlations was significant (see Table 2).

## 4. Discussion

Overall, as the fruit ripened, the cherry fruit color (*CIRG*) deepened, the firmness decreased, and the sugar content increased. Notably, fruits of the “Hongdeng” and “Burlat” cultivars exhibited higher color intensity and sugar content, along with lower firmness. In both the non-choice and choice oviposition tests, *D. suzukii* demonstrated a preference for these two cultivars. The development of *D. suzukii* was significantly influenced by the cherry cultivar, with notable differences observed in the Egg–Larva Period and pupation rate. *Drosophila suzukii* preferred to oviposit on fruits with lower firmness and higher sugar content, where its development was also enhanced. Particularly, the choice oviposition and certain developmental parameters were significantly correlated with the fruit color (*L**, *b**, and *CIRG*). These findings further confirm the importance of the fruit firmness and sugar content in the *D. suzukii* oviposition choice and provide direct evidence of the impact of the fruit color and ripeness on *D. suzukii*’s oviposition preference and fitness.

The findings indicated that *D. suzukii* preferred to oviposit on darker-colored fruits, where its growth and development were also more favorable. Previous research has shown that darker colors, such as black, red, and green, have an attractive effect on *D. suzukii* [17]. In this study, the cultivars with higher oviposition egg counts, including Burlat, Hongdeng, and Tieton, all had red fruits that turned deep red in the overripe stage. However, other studies have indicated that the contrast between fruit and the leaf color, rather than the fruit color alone, might play a more crucial role in orientation and attraction in *D. suzukii* [15,32].

The fruit firmness and sugar content influence the oviposition preferences of *D. suzukii*. Although, in this study, the choice oviposition showed no significant correlation with the fruit firmness or sugar content, a negative correlation with the firmness and a positive correlation with the sugar content were still observed as trends. This aligns with previous studies that found that *D. suzukii* prefers to oviposit on softer-skinned grapes or berries [18,19,33]. Furthermore, as the °Brix level increases, more eggs are laid by *D. suzukii* [34]. The oviposition preference of *D. suzukii* on intact grapes of different cultivars was positively correlated with the glycogen content [35], and similarly, the glycogen content in cherry cultivars was significantly positively correlated with both choice and non-choice oviposition on cherry slices [21].

The primary fruit growth and development factor impacting the growth and development of *D. suzukii* is color. Specifically, darker and deeper fruit colors are associated with higher pupation and hatching rates in *D. suzukii*. We found that fruits with deeper red hues and lower firmness tended to have higher sugar contents. The fruit’s nutritional components, especially the sugar content, affect its suitability for *D. suzukii*. In this study, *D. suzukii* was more likely to complete its full developmental cycle successfully in fruits with higher °Brix levels. As the °Brix levels in blackberries, blueberries, cherries, raspberries, and strawberries increase, more *D. suzukii* emerge as adults [34]. Similarly, there is a significant positive correlation between the °Brix in tart cherries (*Prunus cerasus* L.) and the number of larvae and adults per gram of fruit [36], consistent with our findings. Other research has shown that higher sugar contents in fruits not only shortens the developmental time of *D. suzukii* but also significantly increases the average wing length in adults [19]. However, in our study, while the developmental duration tended to increase with higher °Brix levels, the average developmental periods showed little variation among the cultivars. The egg-to-larva stage ranged from 4.82 to 4.98 d, and the pupal stage ranged from 4.70 to 4.88 d. Further refinement in future studies should focus on more detailed developmental stages to better elucidate the specific effects of fruit physiological traits on growth and development.

As fruit ripens, it becomes softer, darker in color, and sweeter, which makes it more attractive for *D. suzukii* oviposition. Similarly, it has also been found that *D. suzukii* prefers feeding on decaying fruits and tends to lay eggs more readily on ripe fruits [37]. These results provide theoretical support for the commonly held notion that “overripe fruits are more attractive to fruit flies”.

The oviposition choice of *D. suzukii* does not necessarily correlate directly with the fitness of its offspring. In this study, the choice oviposition of *D. suzukii* showed a significant positive correlation with the pupation rate (r = 0.93, *p* = 0.007) and the survival rate (r = 0.97, *p* = 0.002), with no notable correlation with the other developmental parameters. Research has shown that *D. suzukii* prefers to lay eggs on a wild fruit, bird cherry (*Prunus padus* L.), although the adult emergence rates are lower and the development is slower on this fruit, indicating that *D. suzukii* may not effectively select oviposition sites that are more beneficial for offspring development [23]. *D. suzukii* has a wide range of host fruits; in one study, its oviposition preferences on blueberry, lingonberry (*Vaccinium vitis-idaea* L.), chokecherry (*Prunus virginiana* L.), and sea buckthorn (*Hippophae rhamnoides* L.) were compared. Despite severe infestation in blueberry orchards, blueberries were the least preferred fruit in the experiment. This research highlights the opportunistic nature of *D. suzukii*’s oviposition behavior and underscores the critical impact of the fruit phenological stage on the fruit susceptibility [24].

Overall, adult *D. suzukii* showed a clear oviposition preference for the cherry cultivars “Burlat” and “Hongdeng,” displaying higher fitness levels on these two cultivars. Although this study sheds light on the interaction mechanisms between *D. suzukii* and different cherry cultivars, several questions remain for further exploration. For instance, the release of volatile compounds by different cherry cultivars and their specific effects on *D. suzukii* behavior are not yet fully understood. Future studies could delve deeper into these mechanisms to provide stronger scientific support for integrated pest management in the cherry industry. Additionally, considering the potential impacts of ecological factors and climate change on the *D. suzukii* behavior and cherry physiological traits, these factors should also be incorporated into future research to enhance the effectiveness of *D. suzukii* control strategies.

## 5. Conclusions

In conclusion, we compared the physiological traits of the color, sugar content, and firmness among six major cherry cultivars in China and found that *D. suzukii* preferred to oviposit on fruits with darker color, higher sugar contents, and lower firmness. Although the growth and development of *D. suzukii* showed limited variation across the cultivars, significant differences were observed in the egg-to-larvae developmental duration and pupation rates. Moreover, a strong correlation was identified between the cherry color and *D. suzukii* oviposition, as well as certain developmental parameters. These findings elucidate the mechanisms underlying *D. suzukii*’s selection of different cherry cultivars and provide a theoretical basis for developing targeted control strategies and related products.

## Figures and Tables

**Figure 1 insects-15-00984-f001:**
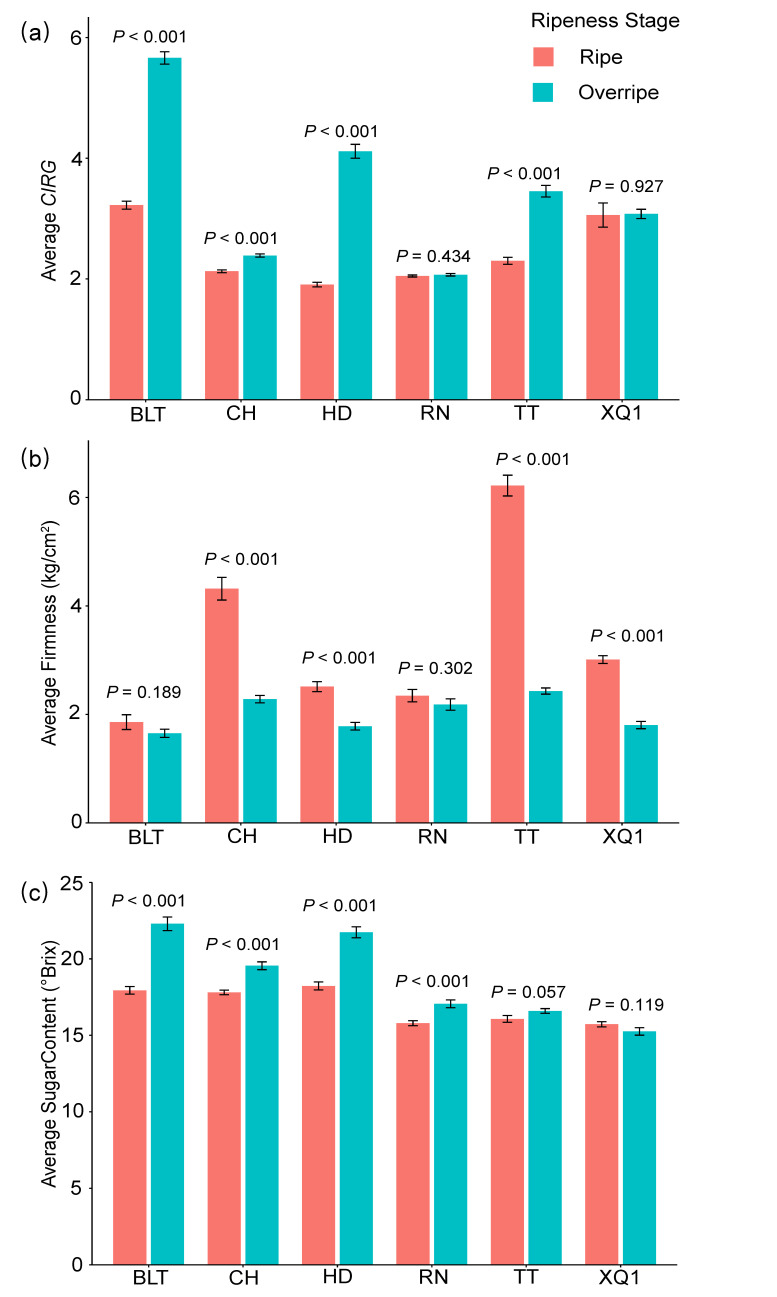
*CIRG* (color index of red grapes), firmness, and sugar content in ripe and overripe stages of different cherry cultivars. (**a**) *CIRG*, (**b**) firmness, (**c**) sugar content. The *p*-values indicated in the figure represent the significance of the differences between the different ripeness stages. *p* < 0.05 indicates a significant difference, while *p* ≥ 0.05 indicates no significant difference. The bar values in the column chart represent standard errors (SDs).

**Figure 2 insects-15-00984-f002:**
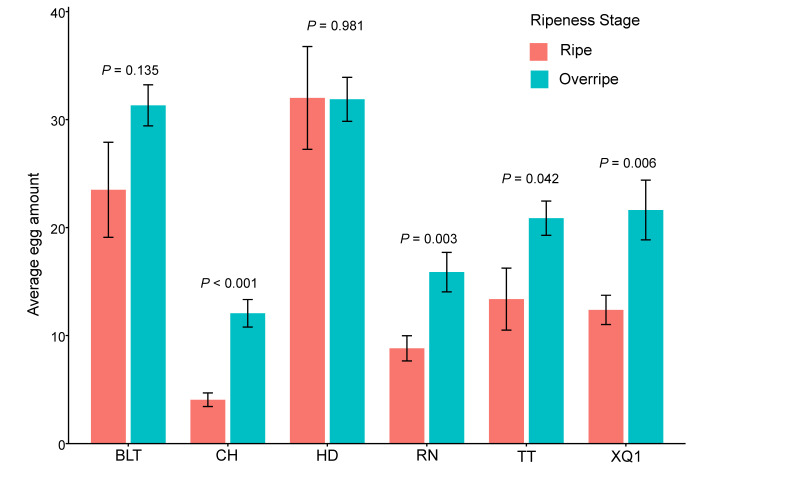
The non-choice oviposition of *Drosophila suzukii* on fruits of different cherry cultivars and at different ripeness stages. The *p*-values indicated in the figure represent the significance of the differences between the different maturity stages. *p* < 0.05 indicates a significant difference, while *p* ≥ 0.05 indicates no significant difference. The bar values in the column chart represent standard errors (SDs).

**Figure 3 insects-15-00984-f003:**
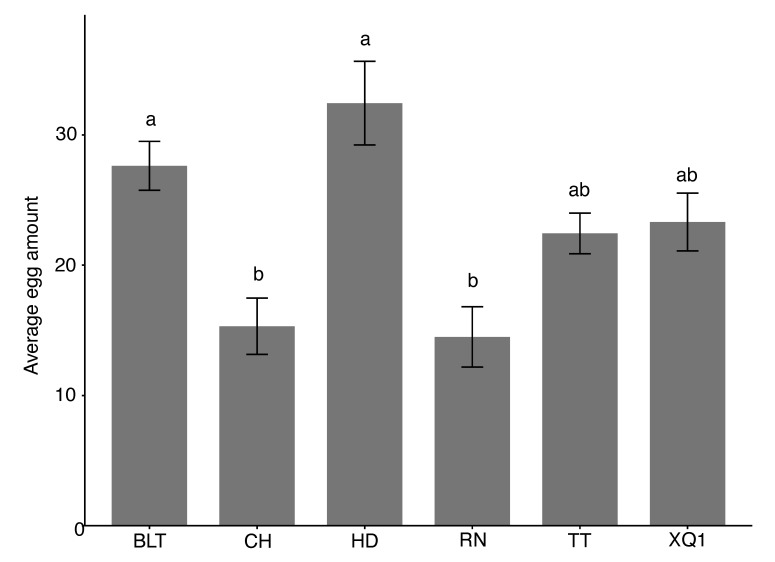
The choice oviposition of *Drosophila suzukii* on fruits of different cherry cultivars during the overripe stage. Different lowercase letters above the bars indicate significant differences, while the same letters indicate no significant differences. The bar values in the column chart represent standard errors (SDs).

**Figure 4 insects-15-00984-f004:**
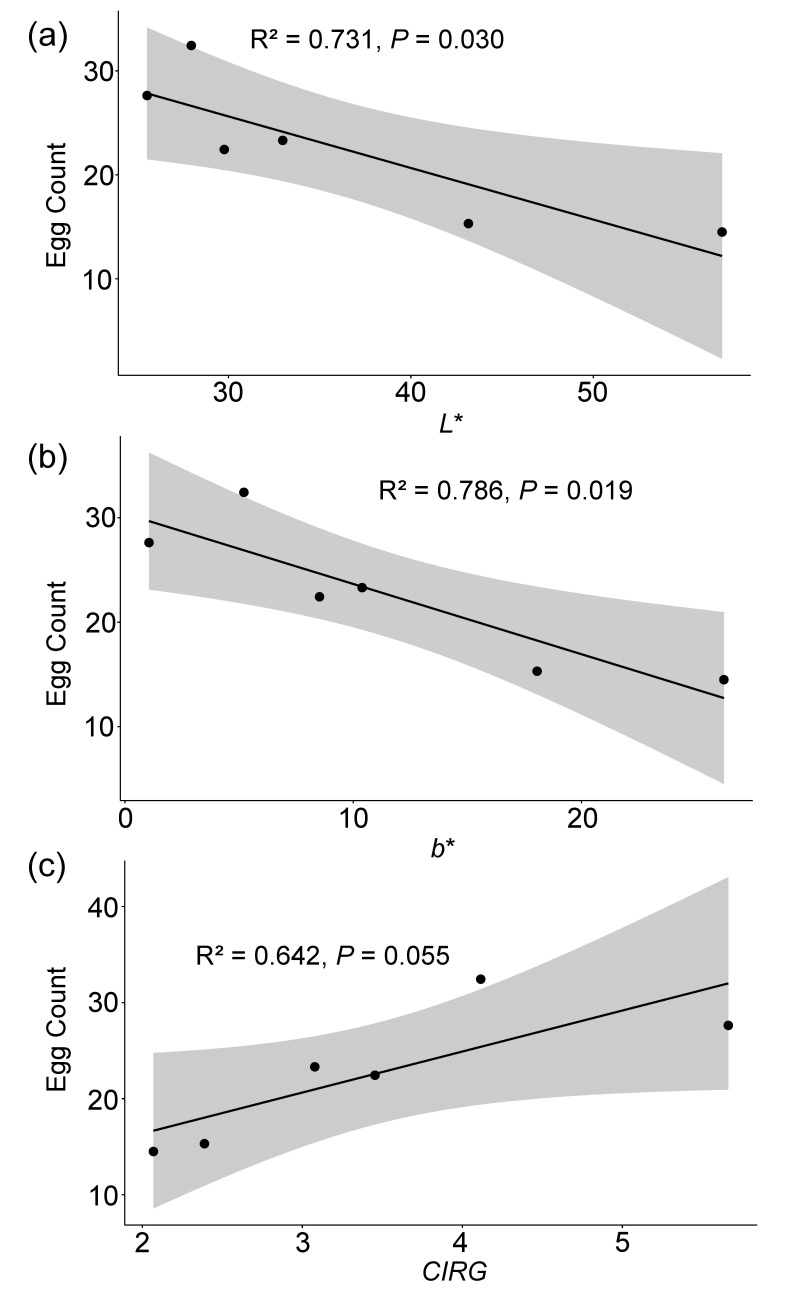
Scatter plots of choice oviposition of *Drosophila suzukii* vs. cherry physiological parameters (*L**, *b**, and *CIRG*) with regression line and confidence interval. (**a**) Egg count vs. *L**, (**b**) egg count vs. *b**, and (**c**) egg count vs. *CIRG*. *L** represents lightness, *b** indicates the yellow–blue axis, and *CIRG* denotes the color index of red grapes. R^2^ represents the proportion of variance explained by the model in the regression analysis. *p*-values indicate the significance levels of the correlations.

**Figure 5 insects-15-00984-f005:**
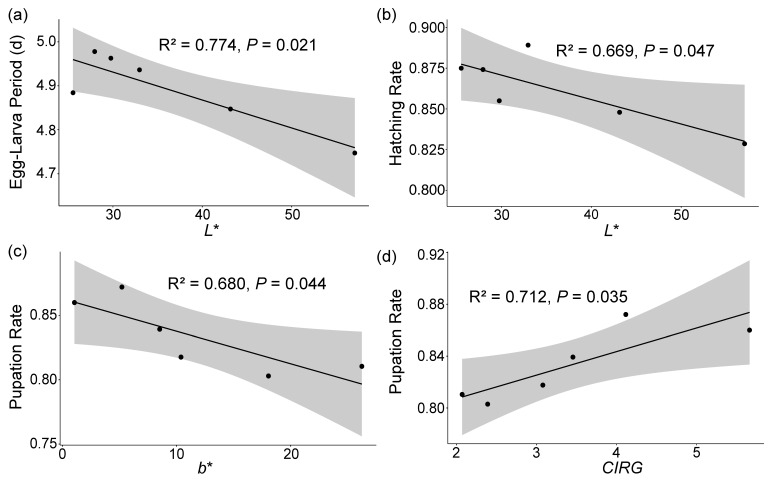
The scatter plot of significant correlations between the developmental parameters of *Drosophila suzukii* and the cherry physiological parameters with the regression line and confidence interval. (**a**) Egg–Larva Period vs. *L**, (**b**) hatching rate vs. *L**, (**c**) pupation rate vs. *b**, (**d**) pupation rate vs. *CIRG*. *L** represents lightness, *b** indicates the yellow–blue axis, and *CIRG* denotes the color index of red grapes. R^2^ represents the proportion of variance explained by the model in the regression analysis. *p*-values indicate the significance levels of the correlations.

**Table 1 insects-15-00984-t001:** Growth and developmental parameters of *Drosophila suzukii* on different cherry cultivars.

Cultivar	Egg–Larva Period (d)	Pupal Period (d)	Female Ratio	Hatching Rate	Pupation Rate	Eclosion Rate	Survival Rate
HD	4.98 ± 0.009 a	4.81 ± 0.038	0.58 ± 0.035	0.87 ± 0.014	0.87 ± 0.014	0.81 ± 0.009	0.62 ± 0.019
TT	4.97 ± 0.012 ab	4.83 ± 0.037	0.67 ± 0.025	0.86 ± 0.009	0.84 ± 0.011	0.74 ± 0.024	0.53 ± 0.022
XQ1	4.95 ± 0.016 ab	4.70 ± 0.033	0.64 ± 0.035	0.89 ± 0.013	0.82 ± 0.020	0.78 ± 0.017	0.57 ± 0.024
CH	4.89 ± 0.023 bc	4.78 ± 0.034	0.64 ± 0.035	0.85 ± 0.012	0.80 ± 0.025	0.75 ± 0.026	0.51 ± 0.030
BLT	4.92 ± 0.020 ab	4.81 ± 0.042	0.59 ± 0.028	0.88 ± 0.019	0.86 ± 0.014	0.76 ± 0.024	0.58 ± 0.032
RN	4.82 ± 0.028 c	4.88 ± 0.026	0.62 ± 0.062	0.83 ± 0.017	0.81 ± 0.015	0.76 ± 0.022	0.51 ± 0.024

Note: Female ratio, hatching rate, pupation rate, and survival rate met normality and homogeneity of variance, and one-way ANOVA followed by Tukey’s HSD post hoc tests were used. Kruskal–Wallis tests and Dunn’s test were applied to the analysis of the Egg–Larva Period, Pupal Period, and eclosion rate. Data are presented as means ± standard errors (SDs). Different lowercase letters indicate significant differences in post hoc comparisons, while the absence of letters indicates no significant differences.

**Table 2 insects-15-00984-t002:** The correlations of choice oviposition, the developmental parameters of *Drosophila suzukii*, and the cherry physiological parameters.

	Egg Count	Egg–Larva Period	Pupal Period	Female Ratio	Hatching Rate	Pupation Rate	Eclosion Rate	Survival Rate	*L**	*a**	*b**	*CIRG*	Firmness	Sugar Content
Egg Count		0.071	0.700	0.204	0.081	0.007	0.119	0.002	0.030	0.559	0.019	0.055	0.126	0.283
Egg–Larva Period	0.774		0.336	0.978	0.087	0.190	0.381	0.119	0.021	0.548	0.059	0.334	0.642	0.844
Pupal Period	−0.203	−0.480		0.792	0.079	0.786	0.528	0.535	0.421	0.229	0.532	0.931	0.486	0.670
Female Ratio	−0.604	−0.015	−0.139		0.588	0.175	0.106	0.125	0.689	0.145	0.508	0.276	0.063	0.073
Hatching Rate	0.758	0.749	−0.761	−0.282		0.312	0.217	0.056	0.047	0.895	0.055	0.196	0.097	0.777
Pupation Rate	0.933 **	0.620	0.144	−0.635	0.500		0.283	0.036	0.079	0.295	0.044	0.035	0.248	0.157
Eclosion Rate	0.703	0.441	−0.327	−0.721	0.591	0.527		0.032	0.513	0.855	0.484	0.619	0.115	0.544
Survival Rate	0.966 **	0.704	−0.321	−0.695	0.801	0.840 *	0.850 *		0.082	0.621	0.061	0.121	0.057	0.311
*L**	−0.855 *	−0.880 *	0.409	0.210	−0.818*	−0.761	−0.338	−0.756		0.932	<0.001	0.045	0.332	0.449
*a**	−0.303	0.311	−0.578	0.671	0.070	−0.515	−0.097	−0.258	0.045		0.678	0.221	0.306	0.256
*b**	−0.886 *	−0.795	0.323	0.341	−0.801	−0.824 *	−0.359	−0.790	0.984 ***	0.218		0.012	0.223	0.321
*CIRG*	0.802	0.481	−0.046	−0.533	0.612	0.844 *	0.260	0.700	−0.821 *	−0.587	−0.908 *		0.141	0.147
Firmness	−0.694	−0.243	0.358	0.788	−0.733	−0.560	−0.709	−0.798	0.483	0.506	0.585	−0.675		0.332
Sugar Content	0.527	0.104	0.223	−0.771	0.150	0.657	0.315	0.501	−0.387	−0.552	−0.492	0.668	−0.483	

Note: Values below the diagonal represent correlation coefficients, while values above the diagonal represent significance levels of the correlations. *L**: lightness; *a**: the red–green axis; *b**: the yellow–blue axis; *CIRG*: the color index of red grapes. *: Significant correlation (*p* < 0.05); **: highly significant correlation (*p* < 0.01); ***: very highly significant correlation (*p* < 0.001).

## Data Availability

The data presented in this study are available within the article and its Appendix A. For additional information or specific requests, please contact the corresponding author.

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
