# Peer review of "Oviposition Preference and Developmental Performance of Drosophila suzukii on Different Cherry Cultivars"

_insects, 2024, doi:10.3390/insects15120984_

Round 1

Reviewer 1 Report

Comments and Suggestions for Authors

The Manuscript insects-3348821 provides relevant biological information on the invasive pest species Drosophlila suzukii. The manuscript is very well written; the abstract is concrete and relevant in information; a background well described; objectives clear and concise; methodology well detailed; data well analysed and results well presented; discussion follows a clear guideline that makes reading easier, although it may be reduced. I find it acceptable with minor revisions. I have made some suggestions that from my view will contribute to its improvement.

Comments:

Page 2, lines 60-62, Authors wrote: --Field experiments revealed that D. suzukii tends to lay eggs on grape cultivars with softer skins, and indoor tests on six types of berries from vineyards also confirmed this preference--.  Where did the field studies take place, in China or in other producing countries?.

Page 2, line 68: Authors wrote: --China cultivates between 30 to 50 major cherry cultivars,…--, does this mean that 30 to 50 charry varieties grow in china?. Whether this is so, it is not clear to me why the authors cannot give the exact number of cherry cultivars growing in China. It would be convenient to make reference to the countries where such field tests were carried out. In addition, the authors should provide information on the importance of cherry crops in other countries, as a comparison with China. Particularly, the cherry varieties which the authors studied and where they are cultivated in other countries or regions of the world, pointing out their economic importance. The latter would highlight even more the good introduction written by the authors.

Page 2, lines 72-82: Aims of the study are clearly stated, as are the motivations for undertaking the study. However, I believe there is a lack of a specific hypothesis of the study, which should be summarized in one or two sentences.

Page 2, lines 86-87: Authors wrote: --The diet formulation for D. suzukii was based on the 86 recipe described by Liu et al. (2019) [22], with minor modifications--. What were these modifications?; please provide a description of them.

Page 4, lines 156-158: Authors wrote: --The number of cultivars used depended on the fruit collection availability on the day (see Table 1, T1-T4)--. I am not clear about this sentence with respect to table 1, because to my understanding table 1 does not show what is referred to the sentence, please analyse this sentence.

Page 4, Table 1: Authors wrote: --Note: Female Ratio, Hatching Rate and Pupation Rate met normality and homogeneity of variance, one-way ANOVA followed by Tukey’s HSD post hoc tests were used. Kruskal-Wallis tests and Dunn’s Test were applied to the analysis of Egg-Larva Period, Pupal Period and Eclosion Rate--. Only the first column Egg-Larva Period has letters to indicate the statistical differences, but the other columns do not; the authors should include letters to identify statistical differences between the different cherry varieties according to evaluated parameters.

Page 9, Table 2: The table is confusing, difficult to understand, too saturated with a lot of information and the data is unstacked. Perhaps, it would be better to restructure the table or to divide the table in parts; it is difficult to understand it.

Page 12, Line 300: The species name must be completely written after a period in a sentence. Therefore --D. suzukii--  must be replaced by Drosophila suzukii.

Page 12, Line 302: Change –Specifically—by –Particularly-.

Page 12, Lines 308-310: The sentences --Previous research had shown that fruit color influenced the attractant response of D. suzukii. Studies found that darker colors, such as black, red, and green, had an attractive effect on D. suzukii [16]-- can be reduced into a single sentence, e.g: -Previous research found that darker colors, such as black, red, and green, had an attractive effect on D. suzukii [16].-

Page 12, Lines 340-341: The sentence --As fruit ripens, it softens, darkens in color, and its sugar content rises, making it increasingly attractive for D. suzukii oviposition-- is confusing, sentence should be restructured.

Page 12, Lines 341-342: The sentence --This study’s findings indicate that D. suzukii prefers feeding on decaying fruits and tends to lay eggs more readily on ripe fruits [32]--, the sentence is confusing because it refers to a published article, rather than to the results of a study undertaken.

Pages 14-15, Reference: some published articles are written in Chinese, they should be written in English.

Comments on the Quality of English Language

Although I am not a native English speaker, I think that the manuscript  needs a birief revision, not a thorough review, as the manuscript is not poorly written.

Reviewer 2 Report

Comments and Suggestions for Authors

The aim of the present study was to reveal correlations between several main parameters of the six major cherry cultivars in China (color, sugar content, firmness, etc) with their preference for oviposition and suitability for development of a serious polyphagous pest, Drosophila suzuki. With this aim the authors conducted a series of simple laboratory experiments. The experiments were well designed and performed. The statistical analysis is correct. The results of the study, although not of high general and fundamental value, can be locally used for the development of optimal methods for the control of this pest. Thus, the manuscript can be published. However, before publication it needs a number of important corrections and improvements (see below).

Lines 20-22: It is a bit strange that this sentence containing abbreviated chromatographic parameters is included in the Simple Summary but not in the Abstract (which is expected to be more complicated and more informative). In the Abstract, the same information is given in much more simple words (lines 32-33). Possibly, this is a mistake?

Line 86: “Over the long term” is not enough for a scientific paper. Please, indicate at least approximated duration of laboratory rearing (in years or in generations) because very long-term rearing in laboratory can have an impact on host preference and suitability.

Line 87: The year when the paper was published is not needed in this journal.

Line 87: Please, describe these modifications because diet also can have an impact on host selection.

Lines 117-118: Please, explain shortly what are chromaticity a* and b* values and give a reference for more detailed description of these parameters.

Table 1: Please, explain either in the head or in the footnote of the table:

1)      The meaning of the letters (significant difference between the values in the same column? p <0.05?, what test was used?, what is the meaning of the absence of the letters?). Although this information is partly given in the Data Analysis section, it should be repeated in the table.

2)      What measure of variation is indicated: SEM or SD? If the measure of variation is the same in all tables, figures, and in the text, this can be indicated in the Methods section. Otherwise, this information should be given for each particular data.

An important general note: please, consider that all figures and tables should be understandable without looking at the text.

Lines 191-193: I would suggest the calculation of an integral parameter, the total pre-adult survival (the number of emerged adults / total number of eggs). Possibly, the differences between the cultivars in this parameter would be clearer and statistically significant.

Line 215: Sorry, but in Fig. 1b just the opposite is clearly seen: as the fruit matured from ripe to overripe stage, its firmness not increased but decreased.

Lines 232, 233, 234 etc.: What measure of variation is indicated in the text: SEM or SD? See also the second comment to Table 1.

Figure 1:

1)      Again, please, explain in the legend what measure of variation is indicated: SEM or SD?

2)      Please, explain in the legend the significance of what difference is shown (I guess, between ripe and overripe fruits, but this should be clearly stated).

3)      Please, show the significance of the pairwise differences between the same ripeness stages of different cultivars – this information is more important than the difference between the two ripeness stages of the same cultivar.

Figure 2: Please, see the comments to Fig. 1.

Figure 3: Please, explain in the legend what measure of variation is indicated: SEM or SD?

Lines 254-255 and 280-282: Significant difference between cultivars in the egg-larval period and in the egg hatching rate is particularly interesting because eggs do not feed. How can you explain these differences? Was something similar recorded earlier with some other fruit or pest species? Please, include this important information in the Discussion.

Line 272: Abbreviations of cherry physiological parameters L*, b*, CIRG should be explained here (in the legends), not only in the Methods section.

Lines 277-279: I guess, a hyphen means not significant correlation but this should be clearly explained. Generally, this table is rather difficult to read and it seems that is was not correctly uploaded to the PDF file. At least for me, it was just impossible to find significance of each coefficient. I would suggest indicating significance not in the other half of the table but somewhere near the corresponding correlation coefficient. Moreover, I would suggest indicating in Table 2 exact data on statistical significance (p = .....) rather than the three levels of significance. Finally, not abbreviations of cherry physiological parameters L*, b*, CIRG but full names should be used in this table.

Line 320: Please, explain here (at least shortly) what is °Brix: to give the reference without any explanations is not enough. Note that nothing is said about this parameter in the corresponding section 2.3.3. “Fruit Sugar Content”. Please, give in this section more detailed explanations about °Brix.

Table S1: Please, explain what data are given. Is it collection dates in “month.day”  format?

Tables S2, S3 and S4: Not only abbreviations of cherry physiological parameters L*, b*, CIRG but full names should be used in all tables.

Tables S3 and S5: What measure of variation is indicated: SEM or SD?

Reviewer 3 Report

Comments and Suggestions for Authors

In this study, Yang et al., examined the egg-laying preferences and developmental parameters of Drosophila suzukii, a major agricultural pest, on six cherry cultivars. The findings show that D. suzukii prefers the Hongdeng and Butlar cultivars for oviposition, correlating with darker fruit color, increased sweetness, and  reduced firmness conditions conducive to fruit fly development. While this study. Based on prior reports of D. suzukii behaviors, its novelty lies in the specific cherry cultivars investigated. Below are comments to improve the manuscript's clarity, organization, and adherence to scientific standards. 

Comments:

1.        Ensure the genus Drosophila is italicized throughout the manuscript for consistency. After the first mention, refer to the species as D. suzukii to avoid redundancy.

2.        Include catalog numbers for all materials and equipment used (e.g., laboratory strains of Drosophila suzukii) to enhance reproducibility.

3.        Expand the figure legends to include sufficient detail about the data presented. This will ensure readers can interpret the figures without referring to the main text. Specifically:

    • Place figure labels (e.g., A, B, C, etc) at the top rather than the bottom.
    • Mention the units for the Y-axis on all graphs.
    • Include the statistical information directly in the figure legend.
    • Check the resolution of the submitted images, which appear to be pixelated. 

  1. Check the statement in Line 214: “As the fruit matured, the cherry fruit’s color (CIRG) deepened, and firmness increased.” This appears to contradict the data presented in the figure 1.

  1. Reorganize Table 2 to improve clarity and readability. Use consistent alignment and formatting. 

  1. The summary and abstract sections currently appear overly similar. Refer to the journal’s instructions (https://www.mdpi.com/journal/insects/instructions)  and revise these sections. 
  2.  
  3. Additional information, if possible image of an assay setup of non-choice and choice oviposition would benefit the readers. 
Comments on the Quality of English Language

There is a scope to enhance presentation, reduce redundancy, and refine sentence structure to improve the manuscript's overall quality and readability.

Round 2

Reviewer 2 Report

Comments and Suggestions for Authors

The authors have addressed all comments. The manuscript was substantially improved. I think that it can be published in the present form.

Reviewer 3 Report

Comments and Suggestions for Authors

In the revised manuscript, the authors satisfactorily addressed the reviewer's comments and improved the presentation.